# The prevalence of AstraZeneca COVID-19 vaccine side effects among Nigist Eleni Mohammed memorial comprehensive specialized hospital health workers. Cross sectional survey

**Mitiku Desalegn**[1☯]**, Gelana Garoma**[2☯]* **, Habtamu Tamrat**[3☯]**, Adane Desta**[4☯]**, Ajay Prakash**[5☯]

1 Department of Anesthesia, College of Medicine and Health Sciences, Wachemo University, Hossana, Ethiopia, 2 Department of Oral and Maxillofacial Surgery, College of Health Sciences, Addis Ababa University, Addis Ababa, Ethiopia, 3 Department of Orthopedics, College of Medicine and Health Sciences, Wachemo University, Hosanna, Ethiopia, 4 Department of Surgery, College of Medicine and Health Sciences, Wachemo University, Hosanna, Ethiopia, 5 Department of Dentistry, College of Medicine and Health Sciences, Wachemo University, Hosanna, Ethiopia

☯ These authors contributed equally to this work.
* gelana.garoma2021@gmail.com

**Data Availability Statement:** All relevant data are within the paper and its Supporting information files.

## Abstract

### Introduction

The best way to eradicate corona virus disease (COVID-19) viral infection is mass vaccination. Many studies demonstrate vaccination is associated with some local and systemic side effects. This study aimed to provide evidence on AstraZeneca COVID-19 vaccine side effects.

### Methodology

Institutional based cross-sectional survey was conducted among 254 health workers at Nigist Eleni Mohammed memorial comprehensive specialized hospital (from July 01/ 2021 to July 30/2021). Data were collected consecutively through self-administered online survey created on Google Forms of platform which had been randomly delivered via (Facebook or telegram pages). Demographic data of participants, side effect after first and second dose of vaccine were covered.

### Result

The prevalence of at least one side effect after first dose was 91.3% and after second dose was 67%. Injection site pain (63.8% vs. 50.4%), headache (48.8% vs. 33.5%), fever (38.8% vs. 20.9%), muscle pain (38.8% vs. 21.7%), fatigue (26% vs. 28.7%, tenderness at the site (27.6% vs. 21.7%), and joint pain (27.6% vs. 20.9%) were the most commonly reported side effects after first and second dose vaccine respectively. Most of participants reported that their symptoms emerged after 6hr of vaccination and only less than 5% of participant's

**Funding:** The authors received no specific funding for this work.

**Competing interests:** No authors have competing interests. The authors have declared that no competing interests exist.

symptoms lasted more than 72hr of post vaccination. The younger age ($\leq$29 year) were more susceptible to at least one side effect ($\chi 2 = 4.2$; $p = 0.04$) after first dose.

## Conclusion

The prevalence of side effect after first and second dose vaccine was higher. Most of the symptoms were short lived and mild. This result might help to solve an emerging public health challenge (vaccine hesitancy) nurtured by misinformation related to vaccines safety.

## Introduction

Corona viruses are single stranded ribonucleic acid (RNA) viruses that cause upper respiratory tract infection [1]. A clinical specimen from a patient having severe acute respiratory syndrome identified a novel coronal virus and named severe acute respiratory syndrome (SARS--CoV-2) [2].

The principal way for transmission of SARS-COVID 19 virus the exposure of the host to respiratory fluid containing the virus primarily Inhalation of air carrying virus, Deposition of virus onto exposed mucous membranes and touching surface exposed to respiratory fluid containing the virus [3].

Pathogenesis of COVID-19 begins when glycoprotein spike on the surface of the virus binds with angiotensin converting enzyme (ACE) receptor of host cell [4]. After binding the viral particle get access to host cell through endocytosis [5]. The fused viral genome carries out a series enzymatic process transported by Golgi vesicles to the cell membrane and released into the extracellular space through exocytosis [6].

Multiple genomic sequence of the virus has made the development of effective vaccine to be limited [7]. 259 vaccine trials are proceeding from November 11, 2020 and the lack of effective vaccine has cost many lives. Several vaccines are developed from numerous trials, from those vaccine one of the vaccine made by AstraZeneca COVID-19 vaccine [8]. COVID-19 Vaccine AstraZeneca is indicated for active immunisation to prevent COVID-19 caused by SARS--CoV-2, in individual's $\geq$18 years old. The Vaccine AstraZeneca is a monovalent vaccine composed of a single recombinant, replication-deficient chimpanzee adenovirus (ChAdOx1) vector encoding the S glycoprotein of SARS-CoV-2. Following administration, the S glycoprotein of SARS-CoV-2 is expressed locally stimulating neutralising antibody and cellular immune responses [9].

The Ethiopian Federal ministry of health has confirmed a coronavirus disease (COVID-19) case on March 13, 2020 in Addis Ababa. The case was announced first to be reported in Ethiopia since the beginning of outbreak in China in December 2019.

COVID-19 Vaccine AstraZeneca has been assessed based on an short-term analysis of pooled data from four on-going randomised, blinded, controlled trials: a Phase I/II Study, COV001, in healthy adults 18 to 55 years of age in the United kingdom (UK); a Phase II/III Study, COV002, in adults $\geq$18 years of age (including the elderly) in the UK; a Phase III Study, COV003, in adults $\geq$18 years of age (including the elderly) in Brazil; and a Phase I/II study, COV005, in adults aged 18 to 65 years of age in South Africa [9].

The vaccination course consists of two separate doses of 0.5 ml each. The second dose should be administered between 4 and 12 weeks after the first dose. Individuals who have taken the first dose of COVID-19 Vaccine AstraZeneca should receive the second dose of the same vaccine to complete the vaccination course. The most frequently reported adverse reactions were injection site tenderness injection site pain, headache, fatigue, myalgia, malaise [10].

Vaccine hesitancy (VH) refers to the "delay in acceptance or refusal of vaccines despite availability of vaccine services"; it is an emerging public health challenge nourished by misinformation related to vaccines effectiveness and safety [11]. This finding was supported in the context of COVID-19 vaccines, because a fear of side effects was the most prominent reason to decrease the readiness of healthcare workers and students in Poland to accept the vaccination [12].

Published data to support adverse reaction of AstraZeneca COVID-19 vaccine are lacking which is a driver of vaccine hesitancy. The knowledge about what happens post vaccination in the actual world among the general population is still modest, thus, by describing what to expect after 1st and 2nd dose of vaccination will help in lowering the apprehension about this type vaccines, increased the public confidence in the vaccines, safety, and accelerates the vaccination process against COVID-19.

The results of this study will be reassuring to those who are fearful of the AstraZeneca COVID-19 vaccine. So, the goal of this study to provide evidence on AstraZeneca COVID-19 vaccine side effects after receiving 1st and 2nd dose of it.

## Materials and methods

An institutional based cross sectional survey was conducted from July 01, 2021 to July 30, 2021 at Nigist Eleni Mohammed memorial comprehensive specialized hospital (NEMMCSH) which is found 230km from capital city of Ethiopia, in Hossana town, Hadiya zone, Ethiopia.

The required data were collected after obtaining ethical clearance from Wachemo University College of medicine and health science institutional review committee. Written informed consent form that included statements about voluntary participation and anonymity was sought from all the respondents prior to data collection. This was accomplished by sending a standardized general invitation letter with the survey link to accept or decline participation to those who took both dose of AstraZeneca COVID-19 vaccine.

The participant who declined consent was not permitted to open the survey and participate in the study, and participants could withdraw from the survey at any time. The members who clicked on the link were directed to the Google forms and to avoid the missing data, the participants will be requested to fill all the questions of the survey or else could not proceed to the next section. No incentives or compensations have been given to participants.

The study employs a self-administered online survey created on Google Forms of platform which had been randomly delivered to NEMMCSH health workers via (Facebook or telegram pages). Potential participants are directed to a page that included brief introduction to the aim and purpose of the study. Data were collected from all who took both dose of the vaccine and sent response during data collection period.

The survey will include two sections, the first section included demographic questions such as (gender, age, profession) second section reviewed the presence of participant's chronic conditions and AstraZeneca COVID-19 vaccine side effects (pain at the vaccination site, tenderness, redness, fever, headache, fatigue, nausea, diarrhoea, muscle pain, back pain). For pilot testing, a questionnaire was passed randomly to 15 participants recently vaccinated and filled the questionnaire after taking the two doses and have been excluded from the study.

The Statistical Package for the Social Sciences (SPSS) version 20.0 was used to carry out descriptive statistics for the demographic variable's similarly, chi square test analysis were performed to assess the correlation between the presence of vaccine side effects and demographic variables. The results were presented by using text, tables, charts and graph.

## Results

### Demographic characteristics of participants

A total of 261 responses were received from respondents. From the total number of responses 7 participants data was incomplete and totally 254 participants were included in the final analysis. 98(38.6%) were females, 156(61.4%) were males and the mean age of the respondents was 29.9±5.8 years old with the median age of 28.5. About 13(5.1%), 68(26.8%), 124(48.8%), 37 (14.6%) and 12(4.7%) were Anaesthetists, Medical doctors, Nurse/Midwife, Pharmacy professional/Lab technicians and Public health experts, respectively. From the total participated health workers, 149(59%) have ≤5 year of work experience and the rest of participants work experience was >5 years (Table 1). Among 254 health workers majority 210 (82.7%) were not tested and 44 (17.3%) tested for COVID-19. Out of 44 health works tested, 19 (43.18%) of them reported positive.

### Prevalence of side effects after first dose vaccine

From the total number of respondents (254), 91.3% (232) participants have reported at least one side effect after first dose of vaccine. Over all, injection site pain was the most prevalent side effect followed by headache (48.8%), fever (38.8%) and muscle pain (38.8%). The prevalence of at least one side effect is slightly greater on males (93.5% vs. 87.7%). At least one side effect among the younger age group (≤29 year old) is nearly greater than participants whose age was >29 year old (94.4%vs 87.7%, respectively) (Table 2).

### Onset and duration of side effects after first dose of vaccine

From the total number of respondents who experienced Side effect, 52.5% of them felt the side effect after 6hr of vaccination and followed by 26.7% (after 1 to 2hr), 18% (3 to 5hr), and 3% (immediately) (Fig 1).

### Prevalence of side effects after second dose vaccine

A total of 69.7% of participants reported to have at least one side effect after second dose of AstraZeneca COVID-19 vaccine. From the rest of side effects, again injection site pain was the most reported symptom with the magnitude of 50.4% and followed by headache 33.5%, fatigue 28.7%, and tenderness at the site 21.7%, fever 20.9% and joint pain 20.9%. There was no

**Table 1. Demographic characteristic of participants who took AstraZeneca COVID-19 vaccine from July 01, 2021 to July 30, 2021 in Nigist Eleni Mohammed memorial comprehensive specialized hospital.**

| Variables | Category | Frequency (%) |
|---|---|---|
| Sex | Female | 98(38.6%) |
| | Male | 156(61.4%) |
| Age | ≤29 year old | 145(57%) |
| | >29 year old | 109(43%) |
| Year of experience | ≤5 year | 149(59%) |
| | >5 year | 105(41%) |
| Profession | Anaesthetist | 13(5.1%) |
| | Medical doctors | 68(26.8%) |
| | Nurse/ Midwife | 124(48.8%), |
| | Pharmacy professionals/Medical Lab technologist | 37(14.6%) |
| | Public health officer | 12(4.7%) |

**Table 2. The prevalence of side effects among health workers after first dose of AstraZeneca COVID-19 vaccine from July 01, 2021 to July 30, 2021 in Nigist Eleni Mohammed memorial comprehensive specialized hospital.**

| Side effects | Category | |
|---|---|---|
| | Yes | No |
| Injection site pain | 162(63.8%) | 92(36.2%) |
| Tenderness at the site | 70(27.6%) | 184(72.4%) |
| Fever | 98(38.6%) | 156(61.4%) |
| Muscle pain | 98(38.6%) | 156(61.4%) |
| Fatigue | 66(26%) | 188(74%) |
| Back pain | 52(20.5%) | 202(79.5%) |
| Joint pain | 70(27.6%) | 184(72.4%) |
| Diarrhoea | 14(5.5%) | 240(94.5%) |
| Headache | 124(48.8%) | 130(51.2%) |
| Nausea | 12(4.7%) | 242(95.3%) |

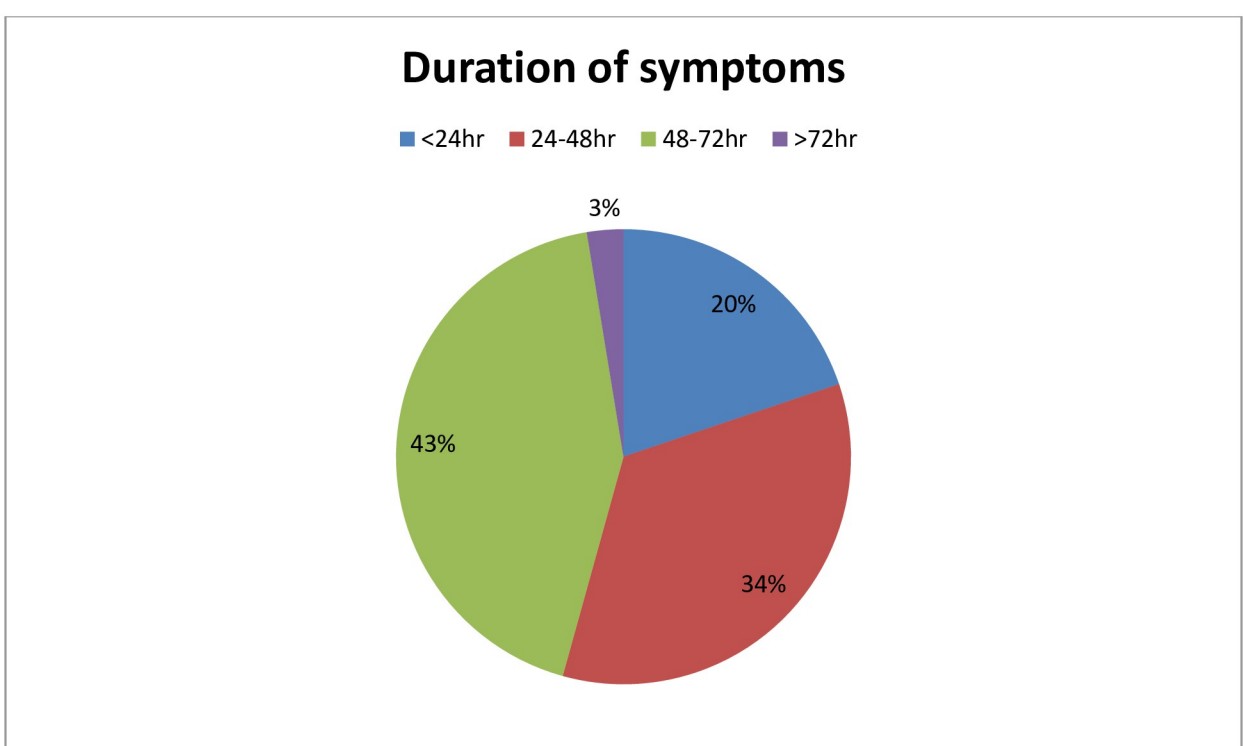

**Fig 1. The duration of side effects of among health workers after first dose of AstraZeneca COVID-19 vaccine from July 01, 2021 to July 30, 2021 in Nigist Eleni Mohammed memorial comprehensive specialized hospital.**

difference on the prevalence of at least one side effect between participants whose age is ≤29 year old and >29 year old (69.7% vs. 69.7%). Regarding sex, there was also no much difference on prevalence of at least one side effect between the two group's male and female (68.5% vs. 71%, respectively) (Table 3).

## Onset and duration of side effects after second dose of vaccine

From the total participants who has experienced at least one side effect, most of emerged after 6hr (39%) of vaccination and followed by 35% (within 1 to 2hr), 14.7% (within 3 to 5hr) and

**Table 3. The prevalence of side effects among health workers after second dose of AstraZeneca COVID-19 vaccine from July 01, 2021 to July 30, 2021 in Nigist Eleni Mohammed memorial comprehensive specialized hospital.**

| Side effects | Category | |
|---|---|---|
| | **Yes** | **No** |
| Injection site pain | 128(50.4%) | 126(40.6%) |
| Tenderness at the site | 55(21.7%) | 199(78.3%) |
| Fever | 53(20.9%) | 201(79.1%) |
| Muscle pain | 55(21.7%) | 199(78.3%) |
| Fatigue | 73(28.7%) | 181(71.3%) |
| Back pain | 52(20.5%) | 202(79.5%) |
| Joint pain | 53(20.9%) | 201(79.1%) |
| Diarrhoea | 14(5.5%) | 240(94.5%) |
| Headache | 85(33.5%) | 169(66.5%) |
| Nausea | 22(8.7%) | 232(91.3%) |

11.3% of them immediately. 54.3% of participants who experience at least one side effect didn't take any treatment measure for the symptoms and about 16.1% of respondents just took bed rest. 29.5% of participants took anti pain to relieve the symptoms (Fig 2).

## Correlation between side effects and participant's age

After first dose of vaccine, the study finding reveals there is significant difference (p = 0.04) between those who were under the age of 29 years and suffering from COVID-19 vaccine side

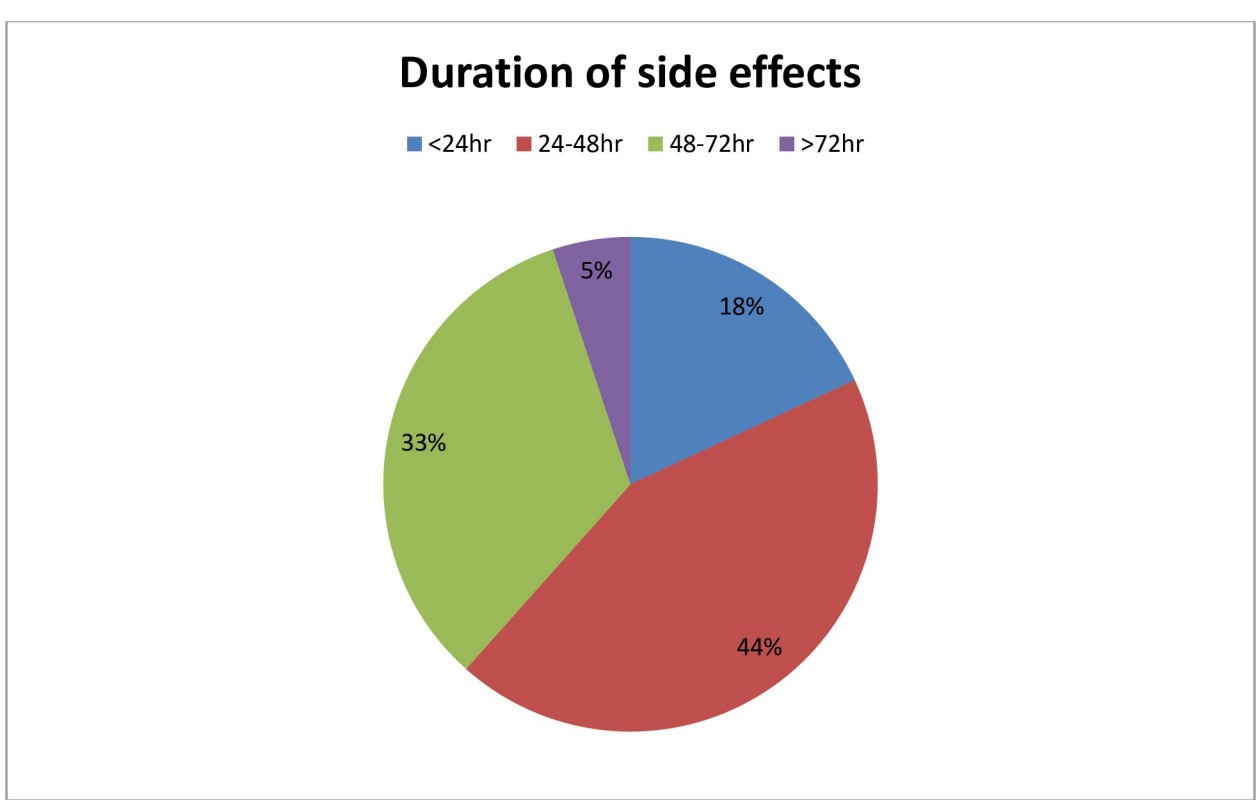

**Fig 2. The duration of side effects of among health workers after second dose of AstraZeneca COVID-19 vaccine from July 01, 2021 to July 30, 2021 in Nigist Eleni Mohammed memorial comprehensive specialized hospital.**

effects and those over the age of 29. There was no significant difference between the two groups (Age ≤29 vs. >29) on side effect reported after second dose of vaccine (Table 4).

## Correlation between side effects and participant's sex

The study result demonstrates there were no significant differences in the number of female participants who reported side effects compared to males after both first and second dose of vaccine (Table 5).

## Discussion

Most of the studies assessed the adverse reaction of Pfizer, Moderna and BioNTech vaccines. There were no sufficient published studies done on side effect of AstraZeneca COVID-19 vaccine. The first shipment of the AstraZeneca vaccines produced by Serum Institute of India (SII) arrived in Ethiopia on 6 March 2021. AstraZeneca was used in our study area among health professionals. Last updated December 2021, three vaccines approved for use in Ethiopia; Oxford/AstraZeneca, serum institute of India/ Covidshield and Sinopharm/ Vero cells. AstraZeneca is the widely used in many setups and the first being available in Ethiopia.

Over all the finding of this study demonstrates the side effects of this vaccine appear to be mild. According to this study more than 90% of respondents have experienced side effect during the first shot. The prevalence of side effect during the second shot of vaccine was lower than the first dose (69.7%), none of this symptoms are serious in nature and requires hospitalization. This result is in line with the cross-sectional survey-based study among German healthcare workers, the frequency of experiencing at least one side effect were 88.1% [13]. Another study conducted in India, 65.9% of respondents reported at least one post-vaccination symptom [14]. A cross sectional survey conducted on residents of Poland shows, among those vaccinated with the first dose of the AstraZeneca vaccine, 96.5% reported at least one post-vaccination reaction. 17.1% of respondents reported all the side effects listed in the survey [15]. The variation in prevalence might be related with unequal sample size or difference in demographic distribution.

According to our study finding, injection site pain was the most prevalent side effect during both first and second dose of vaccine (63.8% vs. 50.4%) and followed by headache (48.8% vs 33.5%), fever (38.8% vs. 20.9%), muscle pain (38.8% vs 21.7%), fatigue (26% vs. 28.7%, tenderness at the site (27.6% vs. 21.7%), and joint pain (27.6% vs. 20.9%). Injection of drug at contracted muscle leads to pain at the site. Injection site pain was reported by many studies to be the most frequent side effect of post vaccination. Cross sectional survey conducted in Czech Republic health workers demonstrates 89.8% of participants reported to have injection site pain and followed by fatigue (62.2%), headache (45.6%), muscle pain (37.1%), and chills (33.9%) [16]. Another study conducted on Saud Arabian inhabitant also reported the short term side effect after first and second dose of COVID-19vaccine. According to this study the

**Table 4. The correlation of participant's age and side effect after first and second dose of AstraZeneca COVID-19 vaccine from July 01, 2021 to July 30, 2021 in Nigist Eleni Mohammed memorial comprehensive specialized hospital.**

|  | Frequency (%) | | Chi-square |
| --- | --- | --- | --- |
|  | Age ≤29(year) (n = 145) | Age >29(year) (n = 109) | P value |
| Side effect after 1st dose | 137(94.4%) | 95(87.7%) | 0.04 |
| Side effect after 2nd dose | 101(69.7%) | 76(69.7%) | 0.999 |

Chi-squared test were used with a significance level of <0.05.

**Table 5. The correlation of participant's sex and side effect after first and second dose of AstraZeneca COVID-19 vaccine from July 01, 2021 to July 30, 2021 in Nigist Eleni Mohammed memorial comprehensive specialized hospital.**

|  | Frequency (%) | | Chi-square |
| --- | --- | --- | --- |
|  | Male (n = 156) | Female (n = 98) | P value |
| Side effect after 1st dose | 146(93.5%) | 86(87.7%) | 0.108 |
| Side effect after 2nd dose | 107(68.5%) | 70(71%) | 0.63 |

Chi-squared test were used with a significance level of <0.05.

most common symptoms were injection site pain, headaches, flu-like symptoms, fever, and tiredness [17].

According to our study, most of respondent's side effects emerged after 6hr of vaccination during both first and second dose of COVID-19 vaccine (52.5% vs. 39%, respectively). Nearly quarter of respondents after first dose and 35% of respondents after second dose reported the onset of symptom was after 1 to 2hr of post vaccination. Regarding the duration of symptoms, most of participants responded their symptoms disappeared with in the first 24 to 48hr of vaccination on both first and second dose of vaccine (44%vs34%, respectively). Only 3% of respondent's symptoms after first dose and 5% of respondent's symptoms after second dose have lasted more than 72hr of post vaccination. This finding is in line with many of studies undergone to assess the side effect of COVID-19vaccine [13–15, 17].

In our study the younger age (≤29 year) were more susceptible to at least one side effect ($\chi^2 = 4.2$; p = 0.04) after first dose of AstraZeneca COVID-19 vaccine. This result is in line with a study done to assess the side effect of COVID-19 vaccine among German health workers [13] and another Cross sectional survey undergone among individuals in UAE [18]. However difference in terms of side effect between male and female were not statistically significant after both first and second dose vaccine.

## Strength and limitation of the study

The finding of this study should be interpreted cautiously regarding the external validity since sex and profession of participants are not equally distributed. The data was collected online through Google form so that only respondents who are motivated will fill and submit the questions which might result for selection bias. Data were collected from health workers who have good understanding about the nature of items, so the outcome were expected to be reported correctly. The data were self-reported which strengthen its objectivity. To the best of our knowledge, this is the first study conducted to assess the side effect of AstraZeneca COVID-19 vaccine among health workers resource limited setting.

## Conclusion

The prevalence of side effect after first and second dose vaccine was higher. Most of the symptoms were short lived, mild and doesn't require hospitalization. This result might help to solve an emerging public health challenge (vaccine hesitancy) nurtured by misinformation related to vaccines safety.

## Supporting information

**S1 Data.**
(XLSX)

## Acknowledgments

This work is dedicated to thousands of fatalities and their families who have fallen victim to COVID-19 in Ethiopia. The authors would also like thank respondents who gave their time to fill and submit the questioner.

## Author Contributions

**Conceptualization:** Mitiku Desalegn, Gelana Garoma, Habtamu Tamrat, Adane Desta.

**Data curation:** Mitiku Desalegn, Gelana Garoma, Habtamu Tamrat, Adane Desta, Ajay Prakash.

**Formal analysis:** Mitiku Desalegn, Gelana Garoma, Adane Desta.

**Methodology:** Mitiku Desalegn, Gelana Garoma, Habtamu Tamrat, Ajay Prakash.

**Supervision:** Ajay Prakash.

**Writing – original draft:** Mitiku Desalegn, Gelana Garoma.

**Writing – review & editing:** Mitiku Desalegn, Gelana Garoma, Ajay Prakash.

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
