## [Decision Letter · Decision Letter 0]

3 Dec 2021

PONE-D-21-34649The prevalence of Astrazeneca covid 19 vaccine side effects among Nigist Eleni Mohammed memorial comprehensive specialized hospital health workers. Cross sectional survey.PLOS ONE

Dear Dr. Garoma,

Thank you for submitting your manuscript to PLOS ONE. After careful consideration, we feel that it has merit but does not fully meet PLOS ONE’s publication criteria as it currently stands. Therefore, we invite you to submit a revised version of the manuscript that addresses the points raised during the review process.

We look forward to receiving your revised manuscript.

Kind regards,

Sanjay Kumar Singh Patel, Ph.D.

Academic Editor

PLOS ONE

Journal Requirements:

(No.The funders had no role in study design, data collection and analysis, decision to publish, or preparation of the manuscript.)

4. Please amend the manuscript submission data (via Edit Submission) to include author Ajay Prakesh.

6. We suggest you thoroughly copyedit your manuscript for language usage, spelling, and grammar. If you do not know anyone who can help you do this, you may wish to consider employing a professional scientific editing service. 

Reviewers' comments:

Reviewer's Responses to Questions

**Comments to the Author**

1. Is the manuscript technically sound, and do the data support the conclusions?

Reviewer #1: Yes

Reviewer #2: Yes

2. Has the statistical analysis been performed appropriately and rigorously? 

Reviewer #1: Yes

Reviewer #2: Yes

3. Have the authors made all data underlying the findings in their manuscript fully available?

Reviewer #1: Yes

Reviewer #2: Yes

4. Is the manuscript presented in an intelligible fashion and written in standard English?

Reviewer #1: Yes

Reviewer #2: Yes

5. Review Comments to the Author

Reviewer #1: The manuscript by Desalegn et al. “The prevalence of Astrazeneca covid 19 vaccine side effects among Nigist Eleni Mohammed memorial comprehensive specialized hospital health workers. Cross sectional survey” is an interesting study about prevalence of Astrazeneca covid 19 vaccine side effects. Authors have conducted a survey on 254 health workers via self-administered online survey created on Google Forms and collected demographic data of participants, side effect after first and second dose of vaccine. Authors have concluded that the prevalence of side effect after first and second dose vaccine was higher. This study is noteworthy and the manuscript requires minor revision before its publication.

Comments

1. Did authors inquire about the COVID-19 infection status of their participants? This will be interesting to know and authors can include this data to their study.

2. At least one additional Figure (illustration) may be provided as to highlight the summary or prospect of this study.

3. The English of manuscript can be polished (minor).

4. The abbreviations should be cross validated in the manuscript (First define them fully followed by abbreviation) and one paragraph can be added for abbreviations.

5. Authors should discuss about the symptoms they found in their study and the reason of these symptoms in discussions.

Reviewer #2: In this paper entitled "The Prevalence of Astrazeneca COVID 19 Vaccine side effects among Nigist Eleni Mohammed memorial comprehensive specialized hospital health worker. cross-sectional survey", the

authors investigate AstraZeneca covid 19 vaccine side effects among health workers. They have done a cross-sectional survey among 254 health workers from July 01, 2021, to July 30, 2021, using google form. The study reported injection site pain, headache, fever, muscle pain, fatigue, and joint pain after the first and second vaccines. However, the symptoms are short-lived and mild. The study is easy to understand and concise. Nevertheless, it requires some minor comments to be answered for publication.

Minor comments:

1) The English may be polished for publication.

2) Please add few lines about the COVID-19 situation in Ethiopia and what COVID-19 vaccines are available other than Astrazeneca in Ethiopia, and if other vaccines are available. Why did the author include only Astrazenca covid-19 in the study?. Is a comparative analysis between the symptoms of different vaccines possible ?.

4) There are many formatting errors in the manuscript and referencing is also not consistent. Please correct it.

5) The author may provide a paragraph regarding challenges or prospects of study in the discussion.

---

## [Author Response · Author response to Decision Letter 0]

22 Feb 2022

Academic editor

1. File naming was edited to comply with the style requirements. We hopefully have no divergences from the style requirements now.

2. The authors received no specific funding for this work. 

3. ORCID Id for corresponding author in Editorial manger included 

4. Author Ajay Prakesh included 

5. reference list reviewed 

6. Language, spelling and grammar was checked online http://Prepostseo.com

Reviewer #1

1. Covid 19 status of our participants inquired and the finding included in result part after the reviewer comment made

2. Concurring comment about one figure about prospective of the study, we made the summary in conclusion part by sentences. If we miss understand it , please more clarification and ready to correct it.

3. English of manuscript was polished for publication 

4. the Abbreviation was cross validated and defined in first use

5. Discussion was made for symptoms following COVID 19 vaccine

Reviewer #2

1. English was polished

2. Few things added about Covid 19 in Ethiopia and the available vaccine in Ethiopia during our study period was only AstraZeneca. So that comparative study was not done

3. We tried to correct formatting errors in manuscript and reference.

4. Challenges of this study separately discussed under limitation of the study

---

## [Editor Report · Decision Letter 1]

24 Feb 2022

The prevalence of Astrazeneca covid 19 vaccine side effects among Nigist Eleni Mohammed memorial comprehensive specialized hospital health workers. Cross sectional survey.

PONE-D-21-34649R1

Dear Dr. Garoma,

We’re pleased to inform you that your manuscript has been judged scientifically suitable for publication and will be formally accepted for publication once it meets all outstanding technical requirements.

Kind regards,

Sanjay Kumar Singh Patel, Ph.D.

Academic Editor

PLOS ONE

---

## [Editor Report · Acceptance letter]

17 Jun 2022

PONE-D-21-34649R1 

The prevalence of AstraZeneca COVID-19 vaccine side effects among Nigist Eleni Mohammed memorial comprehensive specialized hospital health workers. Cross sectional survey. 

Dear Dr. Garoma:

I'm pleased to inform you that your manuscript has been deemed suitable for publication in PLOS ONE. Congratulations! Your manuscript is now with our production department. 

Kind regards, 

on behalf of

Dr. Sanjay Kumar Singh Patel 

%CORR_ED_EDITOR_ROLE%

PLOS ONE